# NoiseGPT: Label Noise Detection and Rectification through Probability Curvature

**Haoyu Wang**
School of Automation
Beijing Institute of Technology
`haoyu.wang@bit.edu.cn`

**Zhuo Huang**
Sydney AI Centre
University of Sydney
`zhuohuang.ai@gmail.com`

**Zhiwei Lin**[*]
School of Automation
Beijing Institute of Technology
`linzhiwei@bit.edu.cn`

**Tongliang Liu**[†]
Sydney AI Centre
University of Sydney
`tongliang.liu@sydney.edu.au`

## Abstract

Machine learning craves high-quality data which is a major bottleneck during realistic deployment, as it takes abundant resources and massive human labor to collect and label data. Unfortunately, label noise where image data mismatches with incorrect label exists ubiquitously in all kinds of datasets, significantly degrading the learning performance of deep networks. Learning with Label Noise (LNL) has been a common strategy for mitigating the influence of noisy labels. However, existing LNL methods either require pertaining using the memorization effect to separate clean data from noisy ones or rely on dataset assumptions that cannot extend to various scenarios. Thanks to the development of Multimodal Large Language Models (MLLMs) which possess massive knowledge and hold In-Context Learning (ICL) ability, this paper proposes NoiseGPT to effectively leverage MLLMs as a knowledge expert for conducting label noise detection and rectification. Specifically, we observe a *probability curvature* effect of MLLMs where clean and noisy examples reside on curvatures with different smoothness, further enabling the detection of label noise. By designing a token-wise Mix-of-Feature (MoF) technique to produce the curvature, we propose an In-Context Discrepancy (ICD) measure to determine the authenticity of an image-label pair. Subsequently, we repeat such a process to find the best matching pairs to complete our label rectification. Through extensive experiments, we carefully demonstrate the effectiveness of NoiseGPT on detecting and cleansing dataset noise, especially on ILSVRC12, the AUROC of NoiseGPT reached over 0.92. And by integrating with existing methods, the classification performance can be significantly improved on noisy datasets, typically by 22.8% on 80% symmetric CIFAR-10 with M-correction. Source code: https://github.com/drunkerWang/NoiseGPT

## 1 Introduction

Contemporary machine learning is greedy for high-quality datasets. However, large-scale datasets such as human-annotated ones like ImageNet [1] and COCO [2] or internet-downloaded ones like WebVision [3] and Instagram Datasets [4] are either resource-consuming or untrustworthy. As a

---

[*]Corresponding author: Zhiwei Lin (linzhiwei@bit.edu.cn)
[†]Corresponding author: Tongliang Liu (tongliang.liu@sydney.edu.au)

38th Conference on Neural Information Processing Systems (NeurIPS 2024).

result, practitioners often have to spend substantial time to conduct prolonged labeling process and the results could still be undesirable. Noise still ubiquitously exist in almost all kinds of datasets. More importantly, dataset noise has been demonstrated to be significantly harmful for training of Deep Learning Models [5]. Consequently, it is urgent to discover efficient and transferable methodology to identify and rectify dataset noise.

To solve this problem, Learning with Noisy Labels (LNL) aims to improve the robustness of Deep Neural Networks (DNNs) through bridging the noise distribution and clean distribution. Existing methodologies [6, 7] commonly leverage loss correction and loss reweighting techniques, where transition matrix [8, 9], for example, is used to capture the noise generation process which enables end-to-end optimization without fitting to noisy labels. Meanwhile, based on the observation that DNNs converge faster on clean examples than noisy ones, sample-selection-based methods [10, 11, 12] divide samples into clean and noisy during training in order to learn from confident examples while exploiting noisy samples [13, 14, 15, 16, 17]. However, it is challenging to estimate accurate noise transition matrix due to the complexity of real-world noise generation process. And the performance of approximation struggles under high-level noise rate without strong dataset assumptions. Consequently, existing methods are largely limited and require to be improved or assisted.

Thanks to the development of MLLMs [18, 19, 20, 21, 22] which have been trained on massive data to effectively fit various real-world data distribution, we propose to leverage MLLMs as knowledge experts to help reduce dataset noise. Based on the empirical findings that noisy data are distributed on different MLLMs probability curvature from clean data, we propose an intuitive hypothesis that MLLMs are inherently optimized for the matching of image-text pairs. Such hypothesis motivates us to propose NoiseGPT which leverages a novel In-Context Discrepancy (ICD) criteria to identify and rectify noisy examples. Particularly, given an example which is in context with its label from dataset, if they match with each other, the MLLM output is stable under perturbation. Conversely, if the image and label are unmatched, the MLLM output will be sensitive to input perturbations. In circumstances where a sample is regarded as noisy, we further leverage CLIP [23] as zero-shot classifier to generate candidate labels. ICD is also applied on these candidates to elect a corrected label with best score. Through extensive studies on datasets such as conventional corrupted versions of CIFAR-10, CIFAR-100 [24], ImageNet ISCVRC2012, as well as real-world grounded datasets like CIFAR-N [25], WebVision [3], the efficacy of NoiseGPT is rigorously validated. As a zero-shot data cleansing method, NoiseGPT demonstrates its powerful ability to scalably distill significantly cleaner versions of noisy datasets than their original ones. Furthermore, our algorithm can be embedded into other LNL methods to further enhance the noisy learning performance under various LNL scenarios. To conclude, our contributions in this paper are as follows:

- We introduce MLLMs as machine experts to cope with noisy labels for the first time, potentially mitigating the reliance on human labor.

- We propose NoiseGPT to tackle the challenge of label noise by leveraging zero-shot capability of MLLMs. We employ a novel In-Context Discrepancy (ICD) criteria combined with the token-wise Mixture-of-Feature (MoF) technique to quantify the possibility discrepancy and identify noisy samples. Additionally, these noisy samples are recycled after label rectification by comparing ICD scores among candidate labels generated by zero-shot classifier CLIP.

- We conduct intricate investigations to evaluate the effectiveness of NoiseGPT. Furthermore, we integrate NoiseGPT as an auxiliary data cleansing method alongside existing LNL algorithms to validate its performance improvement. Additionally, we conduct performance analysis comprehensively understand the details of NoiseGPT.

## 2 Related works

### 2.1 Learning with noisy labels

Existing LNL methods can be categorized into three types, data cleaning, loss-adjustment based approaches and sample-selection based approaches. Data cleaning endeavors to filter out examples whose labels are likely to be corrupted [26, 27]. Previous works in this branch leverages various methods [28, 29, 30] such as bagging, boosting, K-nearest neighbor, anomaly detection to exclude falselabeled instances. However, these methods tend to over-clean samples that are even true-labeled,

resulting in aggravation of shortage of data in many cases. Tendencies of probability curvatures of DNNs [31, 32, 33] during training are also utilized to filter noisy examples. However, their robustness is strongly correlative to the training setting. Loss-adjustment based approaches focus on modifying the loss items before updating the DNNs, including loss correction and loss reweighting. Based on the fact that DNNs using Cross Entropy(CE) loss are prone to overfit noisy data [5], substantial researches have been conducted to design a robust corrected loss [34, 35] by leveraging transfer learning [36, 37, 38, 39], where noise transition matrix [8, 40, 41, 42] has been utilized. However, the performance of loss correction is highly dependent on the precisely-estimated transition matrix, which is hard under heavy noise and large number of classes. And the correction errors will be accumulated during training. Loss reweighting, on the other hand, aims at attributing weighted importance to examples in a designed training scheme to separate clean and noisy examples [43, 44, 45, 46]. Rectifying vectors [47] are also used to guide the classification network through leveraging information from input of the logits and labels. Nonetheless, this approach requires particular reweighting functions and hyperparameters for different noise type and datasets, limiting its practical implementation.

Another line of work, sample-selection, endeavors to identify true-labeled examples from noisy training datasets during training. Researches [48, 49, 13, 50] distinguish clean examples by their early-stage loss in DNNS utilizing memorization effect [51, 52]. Multi-network learning [53, 54, 11] simultaneously trains an additional network to guide the sample-selection. Mix-up [15] employs a semi-supervised mixture model [55] to separate clean and noisy examples, which is integrated into a multi-network framework in DivideMix [13]. SELF [56] leverages unsupervised loss from unlabeled examples while maintaining a running average model called mean-teacher [57, 58]. Pro-Mix [59] leverages the utilities of clean examples by training balanced and unbiased classifiers in a self-supervised framework on separated sub-datasets. Recently, RoCL [60] combines supervised and semi-supervised learning in a two-stage training strategy to exploit both selected clean examples and relabeled examples. Despite their achievement, existing sample-selection methods are intrinsically linked to specific classification tasks, incapable to explicitly provide cleaned versions of datasets in various scenarios. To cope with this, our work aims to provide a transferable noise reduction paradigm to detect and rectify noisy labels by leveraging the zero-shot capability of MLLMs.

## 2.2 Multi-modal models

Recent years, Large Language Models [61, 62, 63, 64, 65, 66, 67] have been successfully applied across different tasks of natural language processing. Concurrently, the arising of Vision Transformers (ViTs) [68] has significantly advanced the development of visual models. In order to align the representations of image and text, CLIP [23] utilizes unsupervised learning by training separate image and text encoders with a contrastive loss on substantial image-text pairs. Besides, researches also endeavor to augment LLM with pretrained visual models to obtain multi-modal LLMs, known as MLLMs [69, 70, 71]. InstructBLIP [72] focuses on equipping MLLMs with the capacity to follow human-instructions on a natural-language interface. The mPLUG-Owl [73] introduces a modularized multi-modal pretraining paradigm to boost its transferability. MMICL [21] proposes a novel context training scheme, allowing the insertion of image features at any position among input text tokens. Based on the powerful zero-shot in-context capability of MLLMs, we propose NoiseGPT, leveraging extensive knowledge acquired from vast multi-modal training examples, to detect label noise. Subsequently, we utilize the zero-shot classification capability of CLIP for candidate labels and compare their matching rates with the input image to derive a rectified label.

## 3 Methodology

In this section, we propose NoiseGPT for label noise detection and rectification, as shown in Figure 1. Given a classifier and an MLLM model parameterized by $\psi$ and $\theta = \{\theta^{enc}, \theta^{dec}\}$, respectively, and a small clean exemplar dataset $\mathcal{D}^{ex}$ with several examples per category to provide prompt for MLLMs, we take advantage of an intriguing *Probability Curvature* effect of MLLM where clean examples $x^{clean}$ and noisy examples $x^{noisy}$ lead to different prediction discrepancies under perturbation, *e.g.*, $\mathbb{E}_{\tilde{x}_i^{noisy} \sim p(\tilde{x}^{noisy}|x^{noisy})} d(x^{noisy}; \tilde{x}_i^{noisy}) < \mathbb{E}_{\tilde{x}_i^{clean} \sim p(\tilde{x}^{clean}|x^{clean})} d(x^{clean}; \tilde{x}_i^{clean})$, where $d(\cdot; \cdot)$ denotes a novel *In-Context Discrepancy* (ICD) measure and $\tilde{x}$ stands for the perturbation of an example under distribution $p(\tilde{x}|x)$. Based on such an effect, we can successfully detect whether a given image-label example pair $x = \{\mathbf{x}, \tilde{y}\}$ has clean labels, *i.e.*, whether $\tilde{y}$ matches with $\mathbf{x}$. By

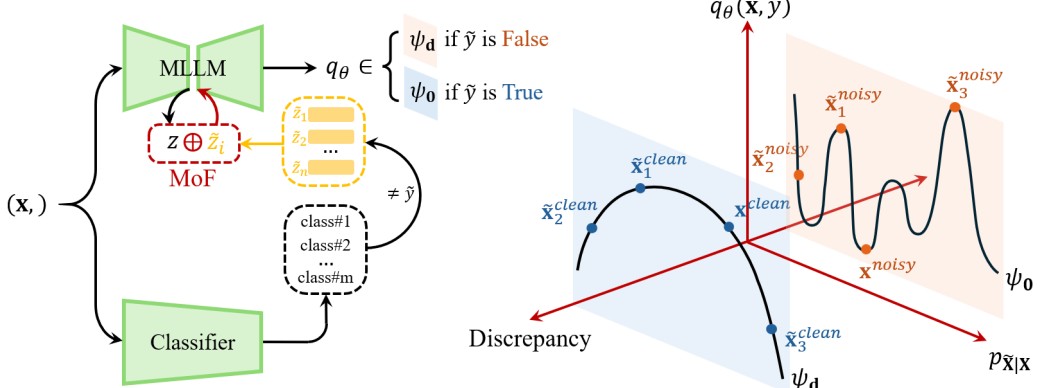

Figure 1: We leverage the zero-shot ability of MLLM to examine whether an example pair is noisy. To identify the potential noise, NoiseGPT first perturbs a given example $x$ and produces a set of augmented versions $\tilde{x}$ via using a novel token-wise Mixture-of-Feature (MoF) technique. Then by comparing the Softmax probabilities $q_\theta$ between $x$ and $\tilde{x}$, we can calculate an In-Context Discrepancy (ICD) measure to further decide the authenticity of the given label $\tilde{y}$.

further exploring the probable class candidates of the classifier, we can effectively find the ground truth label $y$ by choosing the best-matching category. Hence, our NoiseGPT can assist as a dataset cleanser without human intervention.

Generally, NoiseGPT includes two stages: 1) Noise detection and 2) Label rectification. Next, we first demonstrate the noise detection process in Section 3.1, and then we carefully elucidate the details of label rectification in Section 3.2.

## 3.1 Noise Detection

In this section, we propose to conduct noise detection based on the probability curvature effect. Specifically, we observe that clean examples lie on a smooth and convex probability curvature, and noisy examples fall on fluctuated and non-convex curvature, as shown in Figure 2. A similar effect has also been found in Mitchell et al. [74]. As a result, under slight perturbation, the probability value would change differently, which allows us to identify dataset noise. To utilize such an effect, we first conduct a novel token-wise Mixture-of-Feature (MoF) to perturb each example, then, the In-Context Discrepancy (ICD) measure can be calculated to assist detection.

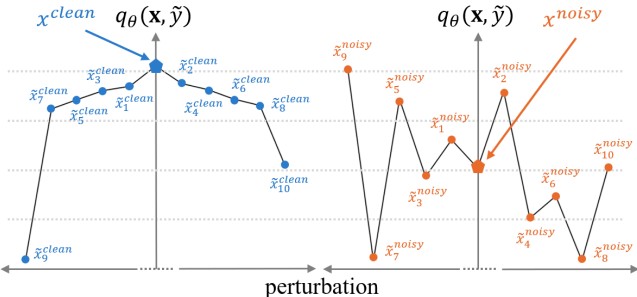

Figure 2: We demonstrate the distinctive curvatures of clean and noisy examples through an experiment. Ten perturbed exemplars are generated for a clean and a noisy sample from CIFAR-100 respectively. The MLLM output Softmax probability $q_\theta(\tilde{\mathbf{x}}|\tilde{y})$ of perturbed clean exemplars $\tilde{\mathbf{x}}^{clean} \sim p(\cdot)$ (**left**) reside within a convex region on the curvature; While those of noisy exemplars $\tilde{\mathbf{x}}^{noisy} \sim p(\cdot)$ (**right**) tend to cluster around the original point, posing lower or higher probability.

**Mixture-of-Feature** aims to perturbate or augment examples at the feature level through interpolation, segmentation, and partial substitution, which avoids changing in the input space as it might cause a mismatch between modalities [75]. Since dataset noise owes to the image example possessing some confusing features that resemble other noisy classes, we propose to mix the features between a given query example $x$ and exemplars $x^{ex} \in \mathcal{D}^{ex}$ from other noisy classes to inject noisy signals. Specifically, our MoF process is formulated as

$$\tilde{\mathbf{z}} = p(\mathbf{z}|\mathbf{z}^{ex}) \triangleq \mathbf{z}[\mathbb{I}(\mathbf{m} = 1)] \oplus \mathbf{z}^{ex}[\mathbb{I}(\mathbf{m} = 0)], \text{ where } p(\mathbf{m} = 1) = w, \tag{1}$$

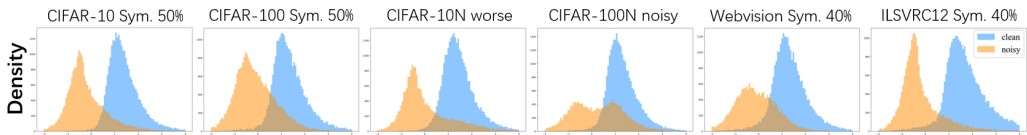

**ICD score under perturbation produced by MLLM**

Figure 3: We conduct validation experiments on six noisy datasets using ICD score, namely CIFAR-10 Sym. 50%,, CIFAR-100 Sym. 50%,, CIFAR-10N with worse labels, CIFAR-100N with noisy labels, Webvision Sym. 40%,, ISLVRC12 Sym. 40%,. We collect scores of 10,000 clean samples and 10,000 noisy samples from each dataset, and each sample is augmented by 10 perturbed exemplars with a MoF weight of 0.5.

where $\mathbf{z}$ and $\mathbf{z}^{ex}$ are the latent representations of input images $\mathbf{x} \in x$ and $\mathbf{x}^{ex} \in x^{ex}$ correspondingly extracted by the vision encoders of MLLMs, $\mathbf{z}[\mathbb{I}(\cdot)]$ denotes the indexing operation that selects the elements from $\mathbf{z}$ where an indexing function $\mathbb{I}(\cdot)$ holds true, $\oplus$ stands for element-wise addition, $p(\cdot)$ stands for the MoF function, and $\mathbf{m}$ indicates a binary mask to select feature elements, which is controlled by probability $w$ to trade off between $\mathbf{z}$ and $\mathbf{z}^{ex}$, larger $w$ reserves more information from the original input in the mixed feature $\tilde{\mathbf{z}}$.

Different from the MoF process conducted using intermediate features in Tong et al. [75] which might break the image patterns further causing misalignment with the subsequent LLM decoder, we propose a token-wise MoF operation that mixes the features by replacing tokens. Intuitively, the latent embeddings $\mathbf{z}$ of MLLMs are sequentialized tokens, each containing specific semantic information. Thus, we conduct MoF from the token level, which can correspondingly perturbate semantic parts across different example pairs. In this way, the mixed feature $\tilde{\mathbf{z}}$ can successfully inherit noisy information and still be aligned with the latent MLLM feature space. After producing mixed features $\tilde{\mathbf{z}}$ as the perturbed versions of original ones $\mathbf{z}$, we combine them with the embeddings of both prompt $s$ and label $\tilde{y}$ and feed them to the MLLM decoder module to obtain the final prediction as $q_{\theta^{dec}}\{s, \mathbf{z}, \tilde{y}\}$. Next, we leverage the In-Context Discrepancy (ICD) criteria to detect dataset noise.

**In-Context Discrepancy** is based on the probability curvature effect where perturbation affects the prediction probabilities of clean and noisy examples differently. Formally, our ICD criteria is calculated as

$$d(s, \mathbf{z}, \tilde{y}; s, \tilde{\mathbf{z}}, \tilde{y}) \triangleq q_{\theta^{dec}}(s, \mathbf{z}, \tilde{y}) - \mathbb{E}_{\tilde{\mathbf{z}} \sim p(\tilde{\mathbf{z}}|\mathbf{z})} q_{\theta^{dec}}(s, \tilde{\mathbf{z}}, \tilde{y}), \tag{2}$$

where $d(\cdot; \cdot)$ denotes the ICD function calculated between two entries, and $p(\tilde{\mathbf{z}}|\mathbf{z})$ is the perturbation distribution for generating $\tilde{\mathbf{z}}$. Intuitively, MLLMs are convexly optimized to associate visual features with corresponding text labels. If an example pair $x$ is clean, *i.e.*, image $\mathbf{x}$ and label $\tilde{y}$ are matched, it will reside in an extrema where its local probability curvature is smooth and convex. On the other hand, if $x$ is a noisy point where the $\mathbf{x}$ and $\tilde{y}$ are mismatched, it might fall on unstable and non-convex curvature.

By injecting a little perturbation as done by the previous MoF process, we observe that the MLLM prediction $q_\theta$ of clean examples will gradually decrease, but for noisy examples, $q_\theta$ would seriously oscillate to a random value around its original one, as depicted in Figure 2. Therefore, our ICD results for clean examples are always positive and relatively larger than noisy ones, further effectively validating the authenticity of input example pairs. To rescale the ICD score of different examples, we further conduct normalization by dividing the results of Eq. 2 by a standard deviation of $q_{\theta^{dec}}(s, \tilde{\mathbf{z}}, \tilde{y})$:

$$\bar{d}(s, \mathbf{z}, \tilde{y}; s, \tilde{\mathbf{z}}, \tilde{y}) \triangleq \frac{q_{\theta^{dec}}(s, \mathbf{z}, \tilde{y}) - \mathbb{E}_{\tilde{\mathbf{z}} \sim p(\tilde{\mathbf{z}}|\mathbf{z})} q_{\theta^{dec}}(s, \tilde{\mathbf{z}}, \tilde{y})}{\sqrt{\mathbb{E}_{\tilde{\mathbf{z}} \sim p(\tilde{\mathbf{z}}|\mathbf{z})} \left[ q_{\theta^{dec}}(s, \tilde{\mathbf{z}}, \tilde{y}) - \mathbb{E}_{\tilde{\mathbf{z}} \sim p(\tilde{\mathbf{z}}|\mathbf{z})} q_{\theta^{dec}}(s, \tilde{\mathbf{z}}, \tilde{y}) \right]^2}}. \tag{3}$$

As a result, we can effectively divide clean examples and noisy ones based on the distribution of $\bar{d}(z, \tilde{z})$ as illustrated in Figure 3. Moreover, the effectiveness of such an effect is carefully validated on various datasets under MMICL [76], a state-of-the-art MLLM that supports effective image-text in-context learning (ICL) in Section 4. In practice, we set a threshold $\tau$ to decide whether an example pair is noisy or not: those with discrepancy scores larger than $\tau$ are considered as clean, otherwise, we further conduct label rectification.

## 3.2 Label Rectification

Thanks to the previous noise detection process, we can effectively validate the authenticity of a given example pair $x = \{\mathbf{x}, \tilde{y}\}$. Moreover, we hope to find potentially correct labels $y$ to rectify the noisy ones $\tilde{y}$. Particularly, we repeat the noise detection process for $C$ probable noisy categories to find the most likely label $y$, formally

$$y \triangleq \arg\max\{\bar{d}(s, \mathbf{z}, \tilde{y}; s, \tilde{\mathbf{z}}, \mathbf{y}_j^{pred})\}_{j=0}^{C}, \quad (4)$$

where $\mathbf{y}^{pred}$ indicates the most probable candidate labels of $\mathbf{x}$ selected by the classifier $\psi$ where the subscript denotes the $j$-th entry. Since various datasets have different numbers of classes, it is infeasible to traverse all classes. Hence, we leverage the predictions of classifier models such as CLIP [23] and select top-$C$ classes as the candidate labels *i.e.*, $\mathbf{y}^{pred} = [\arg\text{sort}(g_\psi(\mathbf{x}))]_{0:C}$ where $[\arg\text{sort}(\cdot)]_{0:C}$ finds the index of top-$C$ elements.

Additionally, to ensure the MLLM output is mapped to a certain probability space in order to provide unified measurement for

---

**Algorithm 1** NoiseGPT: noise identification and rectification.

---
**Input:** sample $x$ and label $y$ from dataset $\mathcal{D}^{set}$, MLLM $q_\theta$, pretrained multi-classifier $q_{cv}$, perturbation function $p$, weight of MoF $w$, number of perturbations $n$, number of candidate labels $C$, threshold $\tau$
1: Uniformly sample $\rho C$ examplers with ground truth labels from $\mathcal{D}^{set}$ to construct a tiny support set $\mathcal{D}^{ex}$;
2: **for** $x$ in $\mathcal{D}^{set}$ **do**
3:     $\mathbf{x}_i \sim p(\mathbf{x}|\tilde{\mathbf{x}}_i^{ex}), i \in [1, n]$       ▷ token-wise MoF
4:     $\tilde{\mu} \leftarrow \frac{1}{n} \sum_{i=1}^{n} q_\theta(\mathbf{x}_i, \tilde{y})$   ▷ approximate expectation in Eq. 2
5:     $d_x \leftarrow q_\theta(\mathbf{x}_i, \tilde{y}) - \tilde{\mu}$
6:     $\tilde{\sigma}_x^2 \leftarrow \frac{1}{n} \sum_{i=1}^{n} (q_\theta(\tilde{\mathbf{x}}_i, \tilde{y}) - \tilde{\mu})^2$
7:     $\bar{d}_x \leftarrow \frac{d_x}{\sqrt{\tilde{\sigma}_x^2}}$             ▷ normalization
8:     **if** $\bar{d}_x > \tau$ **then**
9:         Accept current $\tilde{y}$ as correct label
10:     **else**
11:         $\{y_j^{pred}\}_1^C \leftarrow q_{cv}(x)$   ▷ acquire candidate labels
12:         **for** $j \in 0, 1, \cdots, c$ **do**
13:             $\bar{d}_j \sim \bar{d}(\mathbf{x}, \tilde{y}; \tilde{\mathbf{x}}, y_j^{pred}), j \in [1, C]$
14:             $y \leftarrow \arg\max\{\bar{d}(\mathbf{z}, \tilde{y}; \tilde{\mathbf{z}}, \mathbf{y}_j^{pred})\}_{j=0}^C$
15:         **end for**
16:     **end if**
17: **end for**

---

every example, we conduct prompting to restrict the output to only binary words, *i.e.*, True or False. Moreover, our prompt leverages the ICL ability of MLLMs by providing both correct and incorrect matching exemplars. Our prompt is shown below:

---
Question: This image <IMG_label#$i$> shows a photo of <label#$i$>, True or False? Answer: True;

Question: This image <IMG_label#$j$> shows a photo of <label#$i$>, True or False? Answer: False;

Question: This image <IMG_query> shows a photo of <label#$i$>, True or False? Answer:

---

Specifically, for a query example $x = \{\text{IMG\_query}, \text{label}\#i\}$ whose label prediction from classifier is $i := [\arg\text{sort}(g_\psi(\mathbf{x}))]_0$, we choose one image IMG_label#$i \in \mathcal{D}^{ex}$ to match with label label#$i$ as a True exemplar. Moreover, we choose another image IMG_label#$j \in \mathcal{D}^{ex}$ where $j \in \{0, \cdots, C\}$ and $j \neq i$ as a False exemplar. As shown in Huang et al. [77], such prompt design can effectively teach MLLMs what kind of image-label combination is True or False. As a result, the prediction $q_\theta$ for the query example is based on both the inherent knowledge of MLLMs and the demonstration provided by the prompt. Furthermore, our NoiseGPT restricts $q_\theta$ to binary values which significantly stabilizes the MLLM outputs. As revealed by existing studies [77, 78], providing a set of class candidates and asking which one is the ground truth shows sub-optimal performance. The reason is that current MLLMs cannot effectively conduct multi-class classification and it gets easily confused when facing various choices. Therefore, we only require MLLMs to output binary prediction under demonstrative exemplars which unleashes the power of ICL and benefits making trustworthy inferences. We summarize our methodology in Algorithm 1. Next, we empirically validate the proposed NoiseGPT.

# 4 Experiments

In this section, we first introduce the specifics of our experiment setup. Then we validate the efficacy of our method through an investigation with regard to noise identification and rectification. Furthermore, we undertake a quantitative analysis to demonstrate the enhancing effects of NoiseGPT as data cleansing method through comparing contemporary state-of-the-art LNL models with their combined counterparts with our methodology. Finally, we conduct ablation studies to fully explore the performance of our approach.

## 4.1 Experiments setup

**Datasets** In our experiments, we leverages re-annotated noisy datasets CIFAR-10N and CIFAR-100N [25] which contain real-world human annotation errors. We also generate noisy versions of CIFAR-10, CIFAR-100 [24], WebVision [3] and ImageNet ILSVRC2012 for our studies. The details are as follows:

CIFAR-10N and CIFAR-100N [25]: We adopt Aggregate, Rand1 and Worst versions of CIFAR-10N whose noise rates are 9.03%, 17.23% and 40.21% respectively and Noisy-Fine version of CIFAR-100N whose noise rate is 40.20%.

CIFAR-10 and CIFAR-100 [24]: The CIFAR-10 contains 50,000 labeled images of 10 different classes, while CIFAR-100 contains 100 classes, each with 500 images. We mannually inject 20%, 50%, 80%, 90% symmetric and 40% asymmetric noise into CIFAR-10 and CIFAR-100 respectively.

WebVision [3]: We utilize its validation subset which contains 50,000 images for the 40% symmetric noise condition. Moreover, in order to verify the capacity of NoiseGPT under the real-world circumstance where samples are collected without careful annotation, we utilize mini-Webvision, a subset of Webvision, for noise detection and rectification experiments and test the classification performance on the validation set of Webvision.

ImageNet ILSVRC2012: We utilizes the validation subset which contains 50,000 images and generate symmetric noise for 50% examples in it.

**Models** Primarily, we leverage original CLIP models [23] as our multi-classifier. For MLLM backbone, we employ MMICL [76] which adopts vision encoder of BLIP-2 [70] and FLAN-T5-XXL [79] as the LLM. For comparison with previous LNL works, we consider two methods Pro-Mix [59] and M-correction [35] which train an 18-layer PreAct Resnet for classification tasks.

**Evaluation settings** For exemplars that are used in the in-context learning process, we select 3 images per category to construct a tiny ground-truth support set $\{x_e\}$ to simulate the scarceness of examples in real-world condition. examples are also selected from this support set to generate perturbed sample features. Specifically, for each query sample, we construct $n = 10$ perturbed features with different perturbing resources from $\{x_e\}$. For pseudo labels, we employ the top $C = 3$ prediction of CLIP to conduct label rectification process.

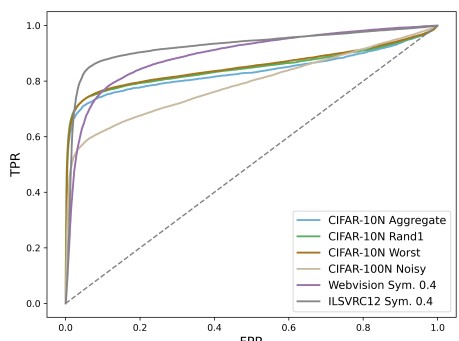

Figure 4: The noise detection ROC curves.

## 4.2 Performance analysis

In this section, we aim to demonstrate the radical capability of NoiseGPT in detecting and rectifying label noise. Our experiments are conducted on 6 datasets: CIFAR-10N Aggregate, Rand1, Worst, CIFAR-100N Noisy, Webvisin Sym. 40% and ILSVRC12 Sym. 40%. The results of classification and rectification are recorded to evaluate the performance.

Table 1: Noise detection and rectification performance.

| Dataset | Noise type | AUROC | Rectification |
|---|---|---|---|
| | Aggregate | 0.8333 | 81.4% |
| CIFAR-10N | Rand1 | 0.8467 | 83.9% |
| | Worst | 0.8508 | 84.9% |
| CIFAR-100N | Noisy | 0.7899 | 59.3% |
| Webvision | Sym. 40% | 0.8935 | 43.2% |
| ILSVRC12 | Sym. 40% | 0.9253 | 43.3% |

**Effectiveness of noise detection** For evaluation metrics, we utilize Area Under the Receiver Operating Characteristic curve (AUROC) to reflect the noise detection performance. AUROC is commonly used to evaluate the performance of binary classification models. Generally, a higher AUROC score (closer to 1) indicates better discrimination ability, while 0.5 suggests random guessing. For label rectification performance, we compare corrected labels of query examples with their true labels to obtain a rectification accuracy.

Table 1 shows the performance of NoiseGPT in noise detection and rectification, the first column shows AUROC scores and the second shows the label correction accuracy. Figure 4 shows the AUROC curves. Our NoiseGPT achieves over 0.83 and 0.78 AUROC score on CIFAR-10N and CIFAR-100N datasets. Especially for Webvision and ILSVRC12, their score reaches over 0.89, which demonstrates the efficacy of filtering out noisy examples. In terms of rectification, NoiseGPT achieves over 80% accuracy on CIFAR-10N datasets. To sum up, NoiseGPT is evident to detect label noise and further recycle most of the noisy examples by rectifying noisy labels to ensure the dataset quantity and quality.

**Comparison with baselines** We compare the detection performance with the baselines in a binary-classification manner with evaluation metrics such as Precision, Recall rate and F1 score with baseline methods. Experiments are conducted on CIFAR-10 sym. 80% dataset, where instances are categorized into 4 types: true-clean, false-clean, true-noisy, and false-

Table 2: Detection performance comparison.

| Method | Precision | Recall | F1 |
|---|---|---|---|
| DivideMix* | 94.36% | 92.63% | 93.49% |
| Proto-Mix* | 96.59% | 93.78% | 94.57% |
| *NoiseGPT* | **97.80%** | **94.39%** | **96.07%** |

noisy. Table 2 shows the results. Note that the detection scores of baselines are from their own detection modules. And the hyperparameters of NoiseGPT are fine-tuned to get the best scores.

## 4.3 Quantitative comparison

While NoiseGPT primarily focuses on zero-shot noise detection and rectification, existing research in Learning with Noisy Labels (LNL) has delved into training noise-robust Deep Neural Networks (DNNs) for classification tasks. To effectively showcase the advantages of our approach, we integrate NoiseGPT as a data cleansing method with two LNL baselines, namely Pro-Mix and M-correction, constituting "NoiseGPT+Pro-Mix" and "NoiseGPT+M-correction", respectively. The classification models are trained under the same settings of as specified in papers of baselines.

Table 3: Noise rectification results.

| Dataset | CIFAR-10 | | | | | CIFAR-100 | | | |
|---|---|---|---|---|---|---|---|---|---|
| Noise type | Sym. | | | | Asym. | Sym. | | | |
| Before | 20% | 50% | 80% | 90% | 40% | 20% | 50% | 80% | 90% |
| After | 7.4% | 13.6% | 19.3% | 24.4% | 9.4% | 16.1% | 28.0% | 40.2% | 44.6% |
| NoC | 46279 | 43206 | 40335 | 37807 | 45282 | 41940 | 35985 | 29918 | 27716 |

**Comparison with classic noisy labels** Our experiments are conducted on the CIFAR-10 and CIFAR-100 datasets, considering varying levels of symmetric and asymmetric noise. Table 3 shows the noise reduction effects of NoiseGPT, which is capable of improving the clean proportion within datasets. Note that the last row shows the number of clean examples after rectification. The improvement is particularly substantial for CIFAR-10 datasets with high noise rate. And for more challenging datasets of CIFAR-100, NoiseGPT still keeps its effectiveness across varing noise conditions.

Subsequently, we transfer cleaned datasets into classification training. We compare the performance of "NoiseGPT+" with their baseline counterparts and other representative works in this field. Note that we re-produce the aforementioned Proto-Mix and M-correction. Due to the lack of detailed training setting information proposed in their papers, some of the re-produced results are not as fine-tuned as what in papers. Nonetheless, we conduct experiments of "NoiseGPT+" with same hyperparameters with their baseline counterparts on each dataset. Thus they

Table 5: Classification on Webvision.

| Method | Accuracy |
|---|---|
| F-correction | 61.12 |
| D2L | 62.68 |
| Co-teaching | 63.58 |
| DivideMix | 77.32 |
| *NoiseGPT+DivideMix* | **78.10** |

objectively demonstrate the effect of NoiseGPT. Table 4 shows experimental results. NoiseGPT poses enhancing effects to LNL works in most noise conditions. Especially in high noise levels, the improvement of classification accuracy is increased by over 20% and 14% respectively for M-correction and Pro-Mix in CIFAR-10 Sym. 0.9.

Table 4: Classification accuracy comparisons.

| Dataset | CIFAR-10 | | | | | CIFAR-100 | | | |
|---|---|---|---|---|---|---|---|---|---|
| Noise type | Sym. | | | | Asym. | Sym. | | | |
| Noise level | 20% | 50% | 80% | 90% | 40% | 20% | 50% | 80% | 90% |
| Cross-Entropy | 82.7 | 57.9 | 26.1 | 16.8 | 85.0 | 61.8 | 37.3 | 8.8 | 3.5 |
| F-correction | 83.1 | 59.4 | 26.2 | 18.8 | 87.2 | 61.4 | 37.3 | 9.0 | 3.4 |
| Co-teaching+ | 88.2 | 84.1 | 45.5 | 30.1 | - | 64.1 | 45.3 | 15.5 | 8.8 |
| Mixup | 92.3 | 77.6 | 46.7 | 43.9 | - | 66.0 | 46.6 | 17.6 | 8.1 |
| P-correction | 92.0 | 88.7 | 76.5 | 58.2 | 88.5 | 68.1 | 56.4 | 20.7 | 8.8 |
| Meta-Learning | 92.0 | 88.8 | 76.1 | 58.3 | 89.2 | 67.7 | 58.0 | 40.1 | 14.3 |
| M-correction* | 93.7 | 92.0 | 87.6 | 68.7 | 91.5 | 68.6 | 59.4 | 47.3 | 12.9 |
| *NoiseGPT+M-correction* | 89.2 | 93.7 | **92.6** | **91.5** | **92.8** | 67.0 | 64.3 | 58.1 | 39.4 |
| Pro-Mix* | 95.9 | 94.6 | 83.1 | 75.0 | 80.5 | **79.6** | **74.6** | 55.4 | 28.9 |
| *NoiseGPT+Pro-Mix* | **96.2** | **94.9** | 88.8 | 89.6 | 92.3 | 76.3 | 71.5 | **63.9** | **47.4** |

**Comparison with real-world noisy labels** In the real-world situation, some datasets prevailing recently are collected from the Internet, such as Webvision. Thus, they contain label noises that represent different patterns from the symmetric. In this paper, we utilize mini-Webvision, a smaller subset of Webvision to verify the capacity of NoiseGPT to combat label noises of this kind. Due to the lack of ground-truth annotations, we directly compare the classification performance with baselines. Following previous works [13], we use the Inception-ResNet v2 [80] as the classification backbone. The results in Table 5 demonstrate that NoiseGPT remains effective on real-world noisy datasets like Webvision. The "NoiseGPT+" denotes that the training set is first cleaned by NoiseGPT.

## 4.4 Sensitivity study

**Effect of perturbation number** Since we approximate the expectation in Eq. 2 with a sequence of perturbations, theoretically increasing the perturbation number $n$ will make the normalized ICD score more robust and effective in distinguishing clean and noisy examples. However, there is a marginal effect when $n$ increases to an extent and becomes computationally unworthy. Figure 5 shows the trend of noise rectification performance under a changing hyperparameter $n$ in the attachment. In our experiments, we select the number of perturbations in order to balance the computational cost and performance.

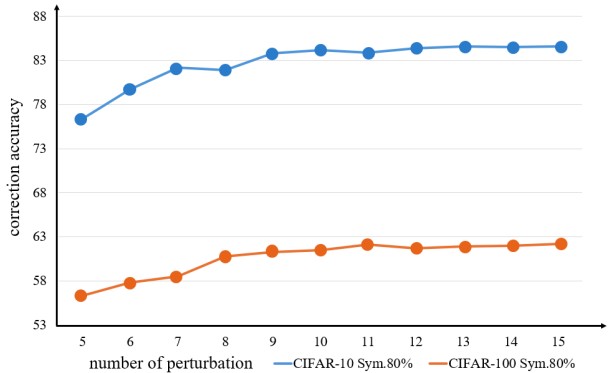

Figure 5: Trend of performance under changing perturbation number.

## 4.5 Ablation study

**The existence of curvatures on different datasets** We further demonstrate the quality of possibility curvature by conducting experiments where query examples from noisy dataset are augmented by a series of perturbed exemplars with varying perturbation strength. According to Section 3.1, we can control the perturbation strength by adjusting token-wise MoF weight $w$, a larger $w$ indicates higher proportion of information coming from query example. We conduct this experiment on four datasets: CIFAR-10 Sym. 50%, CIFAR-100 Sym. 50%, Webvision Sym. 40%, and ILSVRC Sym. 40%.

Figure 6 shows the curvatures of output Softmax probability under changing MoF weights. We calculate the averaged $q_\theta(x)$ of clean and noisy examples in each dataset respectively. It is investigated that on for clean examples, the output Softmax probability $q_\theta^{clean}(x)$ tends to descend as the MoF weight $w$ decreases. Conversely however, the curvature of noisy examples fluctuates optionally

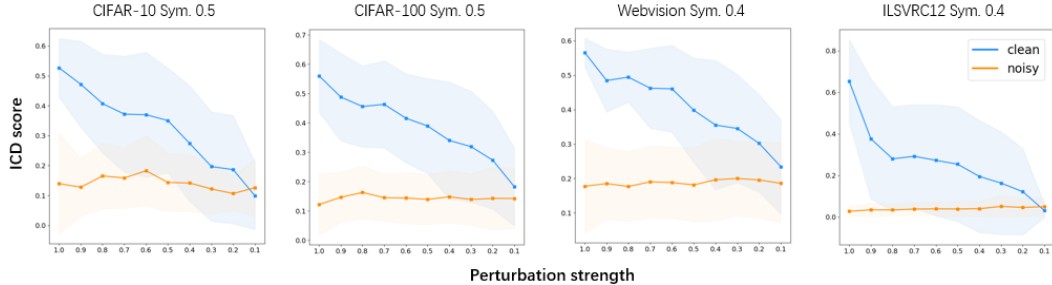

Figure 6: Output possibility curvatures of clean and noisy examples under different perturbation strength.

unconcerned with the change of weight $w$. This phenomenon is confirmed on all datasets, firmly backing up our method, NoiseGPT, which utilizes the discrepancy of output Softmax probability $q_\theta(x)$ under perturbation to distinguish between clean and noisy examples.

**Clean classes that are easier to be considered noisy** Attributing to the fact that clean and noisy examples have different ICD score distributions, our NoiseGPT is capable to detect and rectify noise. However, the ICD score distribution does not remain unchanged among different categories. Some categories of clean examples tend to have relatively higher ICD scores than others, which, in other words, are easier to be mistaken as noisy during the process of NoiseGPT. We investigate such categories

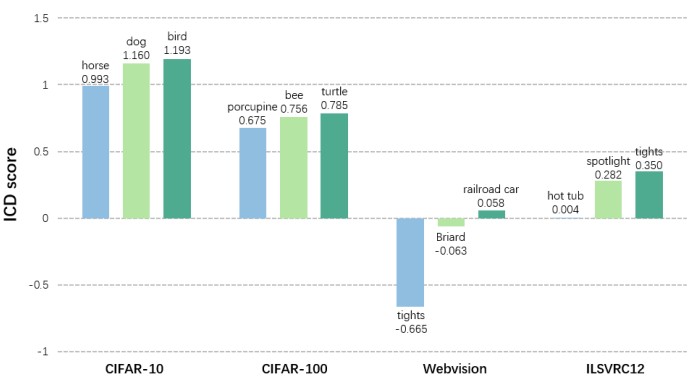

Figure 7: Clean categories that are easier to be mistaken as noisy.

over different datasets, their average ICD scores are recorded as a indication of to what extent they tend to be mistaken. Note that for clean examples, lower score indicates easier to be mistaken. Figure 7 shows which classes are easy to be mistaken for noisy. Further exploration of detection biases are provided in Section 5 in our Appendix.

## 5 Conclusion

**Contribution** In this work, we propose a novel label noise solution via leveraging MLLMs as experts to reduce and recycle noisy instances in datasets. Specifically, we investigate the *probability curvature* of MLLMs under input perturbation. Through a token-wise Mixture-of-Feature technique, we can calculate ICD scores of input examples and divide them into clean and noisy. By conducting extensive quantitative and qualitative experiments on different datasets, our method is validated to sustain effective over varying noise conditions. Moreover, it surpasses previous LNL methods in noise detection and poses substantial potential to cope with other deep learning models to improve their performance. In the future, our method can be further explored for insights into more probelms like OOD detection, weakly-supervised learning, etc.

**Limitation** Despite its adaptability to various datasets and noise levels, the performance of NoiseGPT is constrained by the capabilities of the underlying machine expert it relies on. Research [75] has highlighted the bottleneck effect in vision models within MLLMs. Furthermore, the instruction-following capability of MLLMs significantly influences the distributions of ICD scores, which are closely tied to the confidence of MLLM answers. Enhancing the proficiency of these machine experts can lead to improved performance of NoiseGPT in its tasks.

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

# Appendix

In this section, we first introduce implementations details and computational usage of our experiment. Then, we demonstrate more supplementary experiments to fully explore the qualities of NoiseGPT.

## Implementation details

Among all hyperparameters, MoF weight $w$, number of perturbation exemplars $n$ and number of candidate classes $C$ are of most significance. Theoretically, a larger $n$ avails the approximation in Eq. 2, and for datasets with more complicated classes a larger $C$ is suitable to search for appropriate label. Table 6 shows hyperparameters pf our NoiseGPT experiments. Note that in some of the experiments, we fine-tune the threshold for better noise detection performance.

Table 6: NoiseGPT hyperparameters.

| Hyperparameters | CIFAR-10N |
|---|---|
| MoF weight | 0.5 |
| Number of exemplars per class | 3 |
| Number of perturbations per query | 10 |
| Threshold | 0.7 |
| Number of candidate labels | 3 |

## Compute resources

Our noise detection and rectification experiments of NoiseGPT are powered by GeForce RTX 4090, taking up about 28.5 GiB memory in total for CIFAR datasets. The runtime (hours) of experiments are recorded under $n = 10$, $C = 3$, $w = 0.5$, and illustrated in Table 7.

Table 7: Runtime of NoiseGPT.

| Dataset | CIFAR-10 | | | | | CIFAR-100 | | | |
|---|---|---|---|---|---|---|---|---|---|
| Noise type | Sym. | | | | Asym. | Sym. | | | |
| Noise level | 20% | 50% | 80% | 90% | 40% | 20% | 50% | 80% | 90% |
| Runtime (hour) | 67.5 | 94.2 | 116.4 | 125.8 | 88.3 | 70.2 | 103.7 | 126.5 | 130.9 |

## More classification training performance

In Section 4.3, we have compared the classification performance of "NoiseGPT+" and baselines on CIFAR-10/100 Sym./Asym. datasets. Here we expand this experiment on more datasets: CIFAR-10N Worst and CIFAR-100N Noisy. Table 8 shows the effects of NoiseGPT as a dataset cleansing method. And Table 9 shows the results of classification accuracy.

Table 8: Noise rectification results of CIFAR-N datasets.

| Dataset | CIFAR-10N | | | CIFAR-100N |
|---|---|---|---|---|
| Noise type | Aggregate | Rand1 | Worst | Noisy |
| Noise Rate (before) | 9.03% | 17.23% | 40.21% | 40.20% |
| Noise Rate (after) | 7.92% | 8.07% | 14.84% | 28.45% |
| Clean number | 46038 | 45963 | 42578 | 35774 |

Table 9: Classification accuracy on CIFAR-N datasets.

| Dataset | CIFAR-10N | | | CIFAR-100N |
|---|---|---|---|---|
| Noise type | Aggregate | Rand1 | Worst | Noisy |
| M-correction* | 95.12 | 94.81 | 86.09 | 64.19 |
| *NoiseGPT+M-correction* | 95.16 | 94.79 | 90.36 | 68.20 |
| ProtoMix* | 97.95 | 97.17 | 95.73 | 72.84 |
| *NoiseGPT+ProtoMix* | 97.66 | 97.48 | 96.90 | 73.24 |

## More noise detection and rectification performance

In section 4.2 we introduce AUROC and Label Correction Accuracy as the evaluating metrics for NoiseGPT noise detection and rectification performance. In order to better understand the improving impacts NoiseGPT poses to classification tasks, we further exhibit the evaluating results of NoiseGPT on CIFAR-10/100 Sym./Asym.

Table 10: NoiseGPT performance on CIFAR datasets.

| Dataset | Noise type | AUROC | Correction accuracy |
|---|---|---|---|
| CIFAR-10 | Sym. 20% | 0.9175 | 82.8% |
| | Sym. 50% | 0.9203 | 83.4% |
| | Sym. 80% | 0.9225 | 84.2% |
| | Sym. 90% | 0.9130 | 84.0% |
| | Asym. 40% | 0.8183 | 84.9% |
| CIFAR-100 | Sym. 20% | 0.8949 | 56.4% |
| | Sym. 50% | 0.8969 | 59.9% |
| | Sym. 80% | 0.8903 | 61.5% |
| | Sym. 90% | 0.8848 | 61.9% |

datasets. Table 10 shows the noise detection and rectification results with AUROC score and correction accuracy. Figure 8 shows the ROC curves of noise identification.

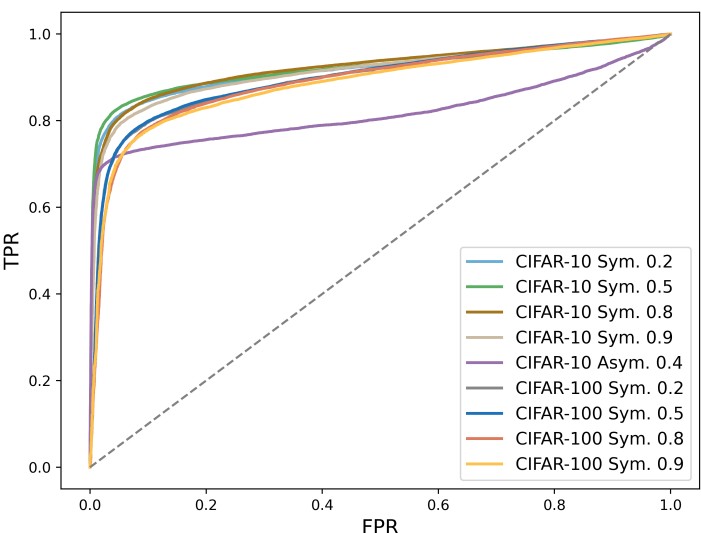

Figure 8: The noise detection ROC curves of CIFAR datasets.

**Noisy classes that are easier to be considered noisy**

In Section 4.5, we illustrate classes that are clean but easier to be mistaken for noisy. Similarly, some categories of noisy examples are easier to be considered clean, obtaining lower ICD scores than their fellows. Contrary to clean classes, higher score for noisy examples indicate that they are easy to be mistaken.

Table 11: Comparison of detection bias.

| Method | 0 | 1 | 2 | 3 | 4 | 5 | 6 | 7 | 8 | 9 | Var |
|---|---|---|---|---|---|---|---|---|---|---|---|
| Proto-Mix* | 9.8 | 11.0 | 9.5 | 7.8 | 10.1 | 8.2 | 10.6 | 11.0 | 10.8 | 11.1 | 1.2 |
| *NoiseGPT* | 10.4 | 10.7 | 9.0 | 10.6 | 10.9 | 9.0 | 10.8 | 8.5 | 10.0 | 10.2 | **0.68** |

**Comparisons on the detection biases with the baseline**

The experiments in Section 4.5 indicate that there are essentially biases in the noise detection stage of NoiseGPT, which will lead to unbalanced example quantities of different classes after rectification. Similar phenomenon also appears in the baselines. Thus, here we compare the biases in noise

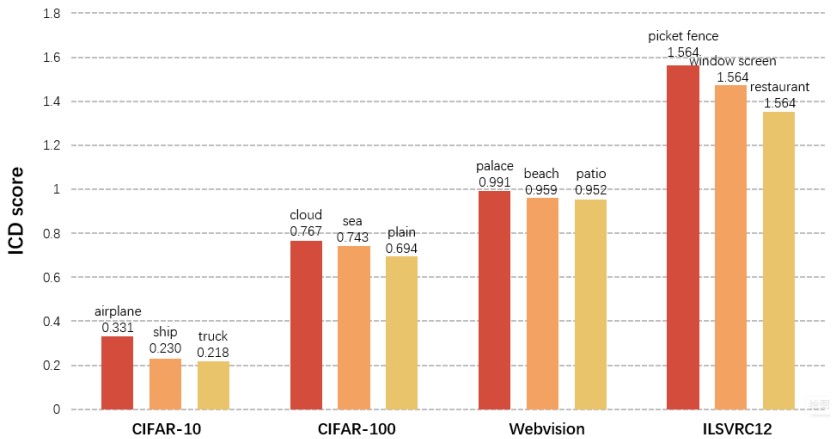

Figure 9: Noisy categories that are easier to be mistaken as clean.

detection between NoiseGPT and Proto-Mix, and calculate the proportions of 10 classes in selected clean data on CIFAR-10 sym. 90%. Table 11 shows the results. Our NoiseGPT exhibits significantly lower variance in the example distributions across different classes, indicating reduced bias in noise detection and rectification compared to Proto-Mix.

