# OpenReview forum: "NoiseGPT: Label Noise Detection and Rectification through Probability Curvature"
_NeurIPS.cc/2024/Conference — NeurIPS 2024 poster_

### Official Review · Reviewer_QCqA · 2024-07-09

**Soundness:** 3
**Presentation:** 2
**Contribution:** 2
**Rating:** 6
**Confidence:** 4

**Summary:**

This paper proposes NoiseGPT, which utilizes a token-wise Mix-of-Feature (MoF) technique and an In-Context Discrepancy (ICD) measure to determine the noisy samples and find best candidate labels with CLIP and MLLM. The effectiveness of this approach is demonstrated through experiments, particularly on the ILSVRC12 dataset, where NoiseGPT achieves an AUROC of over 0.92.

**Strengths:**

1.	This paper utilizes MLLM as expert to detect and rectify noisy labels. Specifically, they propose token-wise Mix-of-Feature and In-Context Discrepancy to detect noise, and use CLIP and MLLM with ICL prompt to find best candidate labels.

2.	Experiments were carried out on various synthetic noise and human annotated noise data sets to demonstrate the performance of NoisyGPT, and further improve the classification accuracy by combining with other LNL methods.

**Weaknesses:**

1.	This paper states that existing LNL methods either require the memorization effect to separate clean data from noisy data or rely on dataset assumptions that cannot extend to various scenarios. However, NoisyGPT seems similar to the former except that it uses LLM as an expert.

2.	This paper needs more ablastion study and baselines to demonstrating the performance of each component of NoiseGPT. For example, what's performance if we directly use CLIP as label corrector in Table 1,2? How does NoiseGPT perform alone in Table 3? What will happen if we use perturbation methods other than MoF?

**Questions:**

•	The paper mentioned they introduce MLLMs as expert to cope with noisy labels for the first time,  is there really no other previous work or recent work？ Can you compare your job with them or some MLLM basic label correction method？

**Limitations:**

•	ICD score distribution does not remain unchanged among different categories. Some categories of clean examples tend to have relatively higher ICD scores than others.

•	Performance of NoiseGPT is constrained by the capabilities of the underlying machine expert it relies on.

---

> ### Author Rebuttal · Authors · 2024-08-07
>
> Thank you for pointing out perspectives that could be further improved.
>
> - Claims of NoiseGPT:
>   - Although there is a lack of works leveraging MLLMs to mitigate label noise, prior researchers have explored its capability in different fields. Huang [1] proposes MVT to supervise vision models using MLLM predictions in OOD scenarios. However, this method cannot be transferred to label noise detection scenarios as the predictions of MLLMs vary ubiquitously among different classes. It is hard to detect and rectify noisy labels from different classes with a unique threshold. Thus, we investigate and leverage the ICD curvature indicating the tendency of MLLM predictions under perturbations to distinguish between clean and noisy examples. On the other hand, CLIP, as a contrastive multi-modal model has been used to select clean examples according to its confidence[2]. And an adaptive loss is constructed for the classifier subsequently through combination with the transition matrix and the class frequency prior to mitigate the overfitting to label noise during training. Nonetheless, NoiseGPT is still distinctive for its way of utilizing pre-trained MLLMs.
>   - About the memorization effects. There are significant differences between the memorization effects [3] that previous works use and the properties of MLLMs we leverage in NoiseGPT. Arpit defines memorization effects as behavior exhibited by DNNs to optimize faster on clean data than noise. Based on the fact that DNNs are prone to produce a higher loss for noise during the early stage of training, existing works [4,5,6] leverage an early stopping strategy or recurrently filter out noisy examples to mitigate the label noise. On the other hand, NoiseGPT leverages the generalization capability of MLLMs in a zero-shot manner. The properties of ICD curvature lie in its inherent optimization to associate visual features with corresponding text labels.
>
>
> - Ablation study:
>   - we add an experiment where different numbers of perturbations are tested. The performance of label rectification is shown in Figure A in the attachment file of the author rebuttal. It is noted that there is a marginal effect when $n$ increases to an extent and it would be unworthy since the computing time grows significantly.
>   - We leverage CLIP ViT-L as a label corrector in comparison with MLLM backbones (including a smaller version of LLM) in Table A to demonstrate the superiority of NoiseGPT.
>   - We tried feature interpolation in the earlier experiments instead of the proposed MoF to introduce noisy perturbation into query examples. The distribution of ICD scores would be different this way, which is illustrated in Figure C in the attachment. We make experiments on CIFAR-10N worse label, which turns out that the overlaps between clean and noisy scores of MoF are smaller compared with that of feature interpolation. Thus, it indicates a better capability to distinguish noisy labels.
>
> [1] Huang Z, Liu C, Dong Y, et al. Machine vision therapy: Multimodal large language models can enhance visual robustness via denoising in-context learning[C]//Forty-first International Conference on Machine Learning. 2023.
> [2] Liang, et al. "Combating Label Noise With A General Surrogate Model For Sample Selection." arXiv preprint arXiv:2310.10463 (2023).
> [3] Arpit D, Jastrzębski S, Ballas N, et al. A closer look at memorization in deep networks[C]//International conference on machine learning. PMLR, 2017: 233-242.
> [4] Song H, Kim M, Park D, et al. How does early stopping help generalization against label noise?[J]. arXiv preprint arXiv:1911.08059, 2019.
> [5] Shen Y, Sanghavi S. Learning with bad training data via iterative trimmed loss minimization[C]//International conference on machine learning. PMLR, 2019: 5739-5748.
> [6] Chen P, Liao B B, Chen G, et al. Understanding and utilizing deep neural networks trained with noisy labels[C]//International conference on machine learning. PMLR, 2019: 1062-1070.

---

> ### Author Response · Authors · 2024-08-13
> **Rebuttal recap**
>
> Dear Reviewer yYJy:
>
> Thank you for your valuable time to review our paper. It has been a while since you last discussed with us. Here we provide a recap to help you read our rebuttal without feeling unfamiliar with this paper.
>
> Your initial concerns can be summarized into two points:
> - The relationship between memorization effects studied in previous LNL works and the ICD in our NoiseGPT.
> - Lack of discussion on previous LNL works using MLLMs as experts.
> - Lack of ablation studies to demonstrate the performance of each component of NoiseGPT and different label corrector backbones.
> - There is a bias in ICD scores among different categories.
>
> Our rebuttal carefully addresses these concerns by:
> - Elaborating on the essential differences between NoiseGPT and previous LNL works.
> - Discussing prior utilization of MLLMs as experts in LNL demonstrating the further insights and superiority of our works, and including them in the related works.
> - Conducting experiments to explore the performance of each component in NoiseGPT, including the influence of hyperparameters $n$ and $\tau$, performance comparison with different backbones (CLIP&FLAN-T5-XL), the results are shown in the PDF attachment of our official author rebuttal.
> - Since other reviewers also put forward concerns about the influence of bias for different categories in label detection and rectification. We conducted another experiment to further compare the biases between NoiseGPT and Pro-Mix. We calculate the proportion each class takes up in selected clean data on CIFAR sym. 90%. The results are illustrated below. We can see that our method shows a much lower variance, which demonstrates the effectiveness and superiority of our method.
> | Method\Class index| 0 | 1 | 2 | 3 | 4 | 5 | 6 | 7 | 8 | 9 | Variance |
> | ----------- | ----------- | ----------- | ----------- | ----------- | ----------- | ----------- | ----------- | ----------- | ----------- |----------- |----------- |
> | Pro-Mix | 9.79% | 11.02% | 9.52% | 7.84% | 10.05% | 8.23% | 10.61% | 11.01% | 10.80% | 11.12% | **1.24** |
> | NoiseGPT | 10.43% | 10.67% | 8.95% | 10.63% | 10.91% | 9.04% | 10.75% | 8.46% | 9.98% | 10.18% | **0.68** |
>
> It is really important to us that you could kindly read our rebuttal and provide further questions if there are any. Thank you so much and hope you have a good day.
>
> Best,
> Authors.

---

### Official Review · Reviewer_81A3 · 2024-07-10

**Soundness:** 2
**Presentation:** 2
**Contribution:** 2
**Rating:** 5
**Confidence:** 4

**Summary:**

This paper proposes to use large multimodal models (LMM) to detect noisy annotations in image datasets. The NoiseGPT method consists in observing whether a controlled perturbation in the latent representation of the LMM leads to a large modification of the LMM response. Experiments in the paper demonstrate that noisy samples are more sensitive to input perturbations than their clean counterparts. This discrepancy is used to detect noisy samples.

Experiments are proposed where NoiseGPT is used in conjunction with existing noise robust algorithms to correct noise on synthetically corrupted datasets as well as the real world CIFAR-N datasets. When NoiseGPT is used, the classification accuracy of datasets is improved.

**Strengths:**

Using LMMs to detect noisy annotations sound next step in using large models for data curation that to my knowledge has not been previously studied.

The observation about the sensitivity of noisy samples to noise in the latent space is intuitive and motivates the approach well.
NoiseGPT going beyond simply prompting the MLM to answer whether the label is clean or noisy strengthens the contribution.

When coupled with existing noise-correction algorithms, the NoiseGPT strategy appears to be beneficial to the test accuracy (Tables 3 and 7).

Some results are proposed on the CIFAR-N datasets in Tables 6 and 7 which hints towards the applicability of NoiseGPT to uncontrolled noisy datasets.

**Weaknesses:**

Although the NoiseGPT idea is relevant to study, I find that the paper severely lacks in insights.
There are no comparison into the possible complementarity of the noise detection between NoiseGPT and ProMix/M-correction. The reader is left to wonder whether NoiseGPT is better in every case or if there exist complementarity in the detection.
There also lacks a baseline comparison/complementarity of the LMM detection with using CLIP directly (as studied in [1] for example) which would be much more efficient to run (see Table 5).

Secondly, because Webvision’s training set is naturally noisy, I believe it would have been more relevant to evaluate the detection capacities of NoiseGPT on these noisy labels instead of artificially corrupting the validation set. This experiment setting goes against previous label noise research evaluating on Webvision [2]. In general, the relevance of presenting ImageNet and Webvision results in this context is limited since both of their validation set are clean (before the proposed synthetic corruption) and classify the same classes.

Finally, because LMMs are trained mostly trained on (and possibly overfit to) noisy data themselves, I would expect that they might struggle to detect noisy samples in uncurrated datasets such as Webvision in its original setting. I believe studying this supposition would be a nice addition to a revised version.

[1] Liang, et al. "Combating Label Noise With A General Surrogate Model For Sample Selection." arXiv preprint arXiv:2310.10463 (2023).

[2] Ortego, et al. "Multi-objective interpolation training for robustness to label noise." Proceedings of the IEEE/CVF Conference on Computer Vision and Pattern Recognition. 2021.

**Questions:**

Why are the experiment settings on Webvision different from previous research ?

Did the authors observe that NoiseGPT struggles to detect web-noisy samples more because they come from the same distribution as the data LMMs are trained on ? (see my last weakness point)

**Limitations:**

NoiseGPT requires a lot more compute than previous approaches to detect noisy samples.
This has been adequately evidenced in the paper (Table 6)

---

> ### Author Rebuttal · Authors · 2024-08-07
>
> Thank you for your suggestions.
>
> - Complementary of noise detection & Baseline of noise detection using CLIP:
>   - To directly compare the effectiveness of our noise detection over the baseline methods such as Pro-Mix and DividMix, we conduct experiments on CIFAR-10 sym. 80% dataset and show the results in Table B in the attachment of the author rebuttal. The compared detection scores are from noise detection modules of baselines. Note that the hyperparameters of NoiseGPT are fine-tuned to get the best scores, which validates the effectiveness of our NoiselGPT.
>   - Moreover, we have made combination experiments on baselines more than M-correction and Pro-Mix, including DivideMix, Cross-Entropy, etc. Experiments show that all the baseline methods perform better after using cleaner datasets produced by NoiseGPT in most cases, verifying the enhancing effect of NoiseGPT. In this paper, we show two combinations that are assumed effective demonstrations of this property.
>   - Meanwhile, there exist cases where complementarity between NoiseGPT and baselines is tested. In fact, MLLM tends to rectify some classes in a dataset more than others, as illustrated in Fig. 6 and Fig. 8. This bias leads to the unbalance in cleaned datasets produced by NoiseGPT, and degrades the performance of classification models in cases like CIFAR-100 sym. 0.2&0.5. To this end, we propose that adjust the threshold τ to prevent overconfident rectifications on clean examples. Results of this method are shown in Figure D in the attachment.
>   - To better demonstrate the efficacy of NoiseGPT, we add an experiment where CLIP ViT-L is leveraged as a label rectifier to assign labels to examples considered noisy. The results are shown in Table A in the attachment in comparison with NoiseGPT on noisy CIFAR datasets.
>
> - Experimental setting Webvision:
>   - Although it would be more relevant to use the training set of Webvision which contains label noise crawled from the Internet, it would be difficult to analyze the performance of detection and rectification without ground-truth labels as reference. Thus, we leverage WebVision to demonstrate the scalability of NoiseGPT over larger datasets with more classes.
>   - For the same reason, there has been hard evidence in experiments indicating diminished detection performance of MLLMs on these datasets.
>   - To demonstrate the capability of NoiseGPT to combat noise web datasets, we visualize the distribution of ICD scores on 20000 examples selected from mini-WebVision[1], which is illustrated in Figure B in the attachment. The overall distribution is similar to what is shown in Figure 3, hopefully indicating the separation of clean and noisy examples.
>
> [1] J. Li, R. Socher, and S.C.H. Hoi. DivideMix: Learning with Noisy Labels as Semi-supervised Learning.

---

> > ### Comment · Reviewer_81A3 · 2024-08-08
> >
> > I thank the authors for their response.
> >
> > - Complementary of noise detection & Baseline of noise detection using CLIP:
> >
> > Thank you for providing the detection scores of a CLIP-based model and smaller LMMs. These baselines are enlightening and should be included in the paper.
> >
> > My comments on the complementarity of NoiseGPT with existing approaches are more directed towards a study of the noise detection biases of LMMs when compared to non-pretrained algorithms such as ProMix or DivideMix. For example, are the difficult classes in Figure 6 and 8 also difficult for ProMix and DivideMix or is this a result of the training data used for the LMM ?
> >
> > - Webvision results
> >
> > Although I agree that detection scores on Webvision are not directly computable, the test accuracy of a model trained with NoiseGPT on Webvision would give an estimation of the capacity NoiseGPT has to remove noisy data harmful to generalization. The separation observed in figure B is however encouraging.

---

> > > ### Author Response · Authors · 2024-08-08
> > > **Further discussion**
> > >
> > > Dear Reviewer 81A3:
> > >
> > > We appreciate your prompt reply,
> > >
> > > - Complementary of noise detection & Baseline of noise detection using CLIP:
> > >   - We will include these comparison results in the later version of the paper.
> > >   - To further compare the bias in noise detection between NoiseGPT and Pro-Mix, we calculate the proportion each class takes up in selected clean data on CIFAR sym. 90%. The results are illustrated below. We can see that our method shows a much lower variance, which demonstrate less bias in noise detection and rectification compared to Pro-Mix.
> > > | Method\Class index| 0 | 1 | 2 | 3 | 4 | 5 | 6 | 7 | 8 | 9 | Variance |
> > > | ----------- | ----------- | ----------- | ----------- | ----------- | ----------- | ----------- | ----------- | ----------- | ----------- |----------- |----------- |
> > > | Pro-Mix | 9.79% | 11.02% | 9.52% | 7.84% | 10.05% | 8.23% | 10.61% | 11.01% | 10.80% | 11.12% | **1.24** |
> > > | NoiseGPT | 10.43% | 10.67% | 8.95% | 10.63% | 10.91% | 9.04% | 10.75% | 8.46% | 9.98% | 10.18% | **0.68** |
> > >
> > >
> > > - Webvision results
> > >   - We make a classification experiment on Webvision using the same framework as Table 3. The “NoiseGPT+” denotes that the training set is first cleaned by NoiseGPT. Note that we train classification models on mini-Webvision and then test on the validation set of Webvision. The results below demonstrate that NoiseGPT remains effective on datasets like Webvision.
> > > | Dataset\Method | ELR | Mix | DivideMix | NoiseGPT+DivideMix |
> > > | ----------- | ----------- | ----------- | ----------- | ----------- |
> > > | Webvision | 73.00 | 74.96 | 77.32 | 78.10 |
> > >
> > > Thanks again for your discussion, we hope to hear your further opinions soon.
> > >
> > > Kind regards,
> > >
> > > Authors.

---

> > > ### Author Response · Authors · 2024-08-12
> > > **Remaining concerns**
> > >
> > > Dear Reviewer 81A3:
> > >
> > > We really appreciate your constructive opinions that helped us improve this paper. We have carefully elaborated on the concerns about:
> > > - Complementary of noise detection. We put forward our comparison experiment results of the noise detection biases existing in NoiseGPT and Pro-Mix, which demonstrates the effectiveness of our method.
> > > - Webvision results. We made a classification performance comparison on Webvision between NoiseGPT and baseline methods.
> > >
> > > If there are any concerns unresolved, we would be glad to have further discussions.
> > >
> > > Thanks again for your time, looking forward to hearing from you soon.
> > >
> > > Best,
> > >
> > > Authors.

---

> > > ### Author Response · Authors · 2024-08-13
> > > **Further discussion**
> > >
> > > Dear Reviewer 81A3:
> > >
> > > We want to thank you again for taking the time to read our rebuttal. We have tried with our maximum effort to address your concerns, but it has been a while since your initial comments. If there are any concerns unresolved, we would be glad to have further discussions.
> > >
> > > Thanks again for your time, looking forward to hearing from you soon.
> > >
> > > Best,
> > >
> > > Authors.

---

### Official Review · Reviewer_gxfG · 2024-07-12

**Soundness:** 3
**Presentation:** 3
**Contribution:** 2
**Rating:** 6
**Confidence:** 4

**Summary:**

The paper introduces NoiseGPT, a method that leverages Multimodal Large Language Models (MLLMs) for label noise detection and rectification in datasets. The approach exploits the probability curvature effect observed in MLLMs, where clean and noisy examples exhibit different curvature smoothness under perturbation. NoiseGPT uses a token-wise Mixture-of-Feature (MoF) technique to generate perturbations and an In-Context Discrepancy (ICD) measure to detect label noise and rectifies them. Experiments demonstrate the method's effectiveness in improving dataset quality and enhancing classification performance in various noisy datasets.

**Strengths:**

1. The paper presents a new application of MLLMs for label noise detection and rectification, introducing the concept of probability curvature and leveraging the zero-shot capabilities of models like CLIP for label rectification.

2. The paper includes experiments across multiple datasets, demonstrating the effectiveness of NoiseGPT in both detecting and rectifying label noise.

3. The paper is well-organized and clearly written. The methodology, including the MoF technique and ICD measure, is explained in detail. The inclusion of algorithm pseudocode and detailed experiment settings enhances reproducibility.

4. The results show significant improvements in label noise detection and rectification, which is crucial for training robust deep learning models. The approach has the potential to be widely adopted

**Weaknesses:**

1. Probability curvature has been previously explored in [1, 2] and [1] specifically for mislabeled detection in vision models, thus reducing the significance of the contribution in the paper, however, one should note [1,2] that these works are limited to vision classification models only. It might warrant discussion in the related work section.

2. It is not clear if the curvature properties leveraged in this paper are restricted to MLLMs, the paper does not provide any evidence for or against it. I.e. do smaller models behave differently or the same? I.e. is a `large' model needed?

3. The paper has missed some similar works, such as [1] Garg et al. (2023) and [2] Ravikumar et al. (2024), which also explore curvature in vision models and mislabelled sample detection. Thus the discussion of these works in the related section should be considered.

4. There needs to be a discussion on the scalability of the method to large-scale datasets, the results in the paper use small subsets of ImageNet and Webvision. Since multiple perturbations are needed on MLLM to detect and correct for label noise, can this method scale to millions and billons of images?



[1] Garg, et al. "Memorization through the lens of curvature of loss function around samples." arXiv preprint arXiv:2307.05831 (2023). \
[2] Ravikumar, et al. "Unveiling privacy, memorization, and input curvature links." arXiv preprint arXiv:2402.18726 (2024).

**Questions:**

1. How sensitive is the method to these hyperparameters? The paper could benefit from an analysis of the sensitivity to hyperparameters like the number of perturbations $n$, and the threshold $\tau$ and more details of how they were chosen.

2. The paper mentions the influence of prompt settings on the reliability of ICD scores. Can the authors elaborate on how different prompt designs impact the performance of NoiseGPT?

3. Is curvature only relevant in MLLM models? I.e. can the baseline be a classification model and do curvature scores in that case perform similarly (see [1]) for mislabeled sample identification?

4. My understanding suggests that the method scales poorly to large datasets since multiple perturbations are needed on MLLM input for rectification. The authors should elaborate on this aspect, discussing potential strategies to mitigate the high computational cost and improve scalability. For instance, are there ways to reduce the number of perturbations required while maintaining detection accuracy? Can the method be parallelized more effectively? Thus the compute requirement and the number of perturbations $n$ and how it affects performance needs discussion.

[1] Garg, et al. "Memorization through the lens of curvature of loss function around samples." arXiv preprint arXiv:2307.05831 (2023).

**Limitations:**

1. The paper adequately addresses some limitations, such as the reliance on MLLMs' vision capabilities and the effect of prompt settings on ICD scores. However, the computational cost associated with the method is not fully discussed. Since NoiseGPT requires generating multiple perturbations (n perturbations) for each example, this can lead to significant computational overhead, particularly when dealing with large datasets. This overhead may result in slow processing times, as the models involved (e.g., CLIP, large language models) are computationally intensive.

2. It is not clear if the curvature properties leveraged in this paper are restricted to MLLMs, the paper does not provide any evidence for or against it. I.e. do smaller models behave differently or the same?

---

> ### Author Rebuttal · Authors · 2024-08-07
>
> Thank you for your suggestions.
>
> - Relationship to previous loss curvature works:
>   - The curvature of loss function is a common phenomenon in deep networks, existing works have studied in some fields but none of them have been conducted in LNL. Our work is the first to discover the effectiveness of learning curvature to benefit LNL, which can establish our contribution and novelty in this field.
>   - Moreover, existing curvature studies are based on the classification probability, but our ICD measure considers the knowledge of in input prompt by carefully leveraging the in-context learning ability, which can benefit the effectiveness of the MLLM identification and rectification.
>   - Furthermore, our ICD score does not need pre-training to be used, we only need to conduct zero-shot inference to implement this score, which is much more efficient and straightforward.
>
> - Specification of model for curvature properties:
>   - Our framework which leverages the ICL capability can only be implemented using MLLMs. Maybe in the future when vision models start to show ICL ability, we can also implement NoiseGPT to them.
>   - Although the ICD curvature is restricted to MLLMs, smaller models of this kind also share this property, but with a drop in performance. For comparison, we adopt FLAN-T5-XL as another LLM backbone which has a size of 3B parameters and is smaller than FLAN-T5-XXL (11B). The performance of reducing noise is illustrated in Table A in the attachment file of the author rebuttal.
>
> - Missing references and discussions of the related works:
>   - Thanks for pointing them out, we will complement all the mentioned references, and a discussion is added in the related work section in our author rebuttal due to the character limitation here.
>
> - Discussion on the scalability to large-scale datasets:
>   - The computational cost required for perturbation will not increase as the scale of datasets grows. We select a few exemplars from each class in the dataset to construct a subset D^ex at first. All perturbations of subsequent MLLM input query examples are injected from this subset, and no more exemplars are needed when the scale of the dataset grows.
>   - The computing time of the proposed method grows proportionally to the number of examples in a dataset. Thus, when faced with datasets of a much larger scale, we suggest that adjust the hyperparameter $n$ to reduce the number of perturbations for each input example. As the computing time also shrinks proportionally with the decrease of $n$, since $n$ determines how many times MLLM needs to generate an answer to produce ICD scores. By selecting a relatively smaller $n$, the performance of NoiseGPT can be maintained adequately while saving much of computing time.
>
> - Parameter sensitivity analysis & Effect of perturbation numbers and how to reduce it for efficiency:
>   - Threshold $\tau$: To further explore the influence of threshold on the overall noise mitigation, we make experiments on CIFAR-100 sym. 20% and 50% with varying τ. Besides, as $\tau$ only affects the noise detection of NoiseGPT, which is a binary classification scenario. We adopt the ROC curve and AUROC score to better evaluate changes in detection performance along with different thresholds, which reflects the changes in TPR and FPR as the threshold moves from the smallest to the largest. By analysing the ROC curve of a small subset of query examples, a suitable threshold can be inferred for the whole dataset.
>   - Number of perturbations $n$: Theoretically, more perturbations make the normalized ICD score more robust and effective in distinguishing clean and noisy examples. However, there is a marginal effect when $n$ increases to an extent when it becomes computationally unworthy. In Figure A in the attachment, the sensitivity of hyperparameter $n$ is explored. In practice, we select the number of perturbations $n$ by balancing the computational cost and performance.
>
> - Prompt design: We based our prompt settings on previous works [1] and made some modifications, including the sequence and the content.
>   - The proposed prompt includes three sentences, a positive Q&A, a negative Q&A, and a query question whose answer is to be generated. In earlier experiments, we changed the order of the first two sentences which were negative, affirmative, and query. It was observed that MLLM tended to output a lower Softmax possibility of answering “true” for all query examples according to the regularity in former lines of prompts. Thus, we bypass this problem using the ICD score which depends on the relative magnitude of output possibilities between original and perturbated examples, instead of the absolute possibilities of “true” or “false”.
>   - In experiments, we extended the prompts to different sizes in the same format shown in Sec. 3.2. And different sequences of these sentences were also investigated. It turned out that the tendency mentioned in the last paragraph was facilitated.
>   - To completely mitigate the influence of prompts, we assume that the representation editing [2,3] methods could have possibly been a substitution for existing prompt learning methods. We are still working on this idea.
>
> - Baseline using learning curvature with classifier model:
>   - The loss function produced by classification models is related to the discrepancy between the prediction $y’$ and label $y$, which are essentially different from ICD. As a result, it would have been complicated and counterintuitive to investigate in probability changes of DNNs induced by perturbation.
>
> [1] Huang Z, Liu C, Dong Y, et al. Machine vision therapy: Multimodal large language models can enhance visual robustness via denoising in-context learning[C]
> [2] Liu, S., Ye, H., Xing, L. & Zou, J. In-context Vectors: Making In Context Learning More Effective and Controllable Through Latent Space Steering.
> [3] Kong, L. et al. Aligning Large Language Models with Representation Editing: A Control Perspective.

---

> > ### Author Response · Authors · 2024-08-12
> > **Remaining concerns**
> >
> > Dear Reviewer gxfG:
> >
> > We thank you again for your valuable time to review this paper, your constructive advice is really helpful.
> >
> > By carefully answering all your concerns, this paper has been improved on scalability to larger datasets, the sensitivity of hyperparameters, the prompt settings, and the invetigations in different backbone models. We hope to know whether our rebuttal solves your concerns. Since the NeurIPS conference supports interactive discussion, we wish we could have to chance to make further efforts to polish our work.
> >
> > Thanks again for your previous help, we hope to hear from you soon!
> >
> > Best,
> >
> > Authors.

---

> ### Author Response · Authors · 2024-08-11
> **Further discussion**
>
> Dear Reviewer gxfG:
>
> We want to express our appreciation for your valuable suggestions, which greatly helped us improve the quality of this paper. We are also glad that you agreed that our idea is novel and has the potential to be widely used.
>
> We have made our maximum effort to address your concerns on scalability to larger datasets, the sensitivity of hyperparameters, and the unique properties of ICD curvatures in MLLMs, and etc.
>
> Your further opinions are very important for evaluating our revised paper and we are hoping to hear from you. Thank you so much.
>
> Best,
> Authors.

---

### Official Review · Reviewer_TkfP · 2024-07-15

**Soundness:** 3
**Presentation:** 2
**Contribution:** 1
**Rating:** 3
**Confidence:** 4

**Summary:**

This paper employs Multimodal Large Language Models to detect noise samples. The In-Context Discrepancy is utilized to quantify the discrepancy between the original and perturbed samples. Additionally, the identified noise samples are integrated with the Pro-Mix and M-correction noise label learning framework. Furthermore, a comparison between the proposed method and existing approaches is conducted.

**Strengths:**

Strengths: 1) The author utilizes Multimodal Large Language Models to detect noise samples.

**Weaknesses:**

Weakness;
1) The novelty of the approach is limited.
2) Some previous works are not properly cited, such as the lack of citation for In-Context Discrepancy.
3) Several highly relevant works are missing, such as those measuring the In-Context Discrepancy between original and perturbed samples, including recent publications.

[1] Early-learning regularization for preventing memorization of noisy labels.
[2] Strategies for preventing continuous damage caused by noise during model training.
 [3] A survey on learning from noisy labels with deep neural networks.
[4] Learning to rectify for robust learning with noisy labels.
[5] ...

4) The efficacy of the proposed method is questionable, as Table 3 shows only a limited improvement compared to Pro-Mix*.


Some typos:
"Figure ?? "

"50%„ "

**Questions:**

The efficacy of the proposed method is questionable, as Table 3 shows only a limited improvement compared to Pro-Mix*.

The author is recommended to include a section on "Sample cleaning" in the related work to provide a comprehensive coverage of relevant studies.

**Limitations:**

YES.

---

> ### Author Rebuttal · Authors · 2024-08-07
>
> Thanks for your constructive comments.
> - Limited novelty:
>   - We make the first contribution to effectively detect and rectify label noise without the need for pre-training, providing a novel direction for learning with label noise, which has never been previously studied (**Reviewer 81A3**), and has the potential to be widely adopted (**Reviewer gxfG**). We hope the Reviewer could further provide on which aspect he thinks our novelty is limited, and we would be happy to try our best to address them.
>
> - Missing previous works & several relevant works.
>   - We will carefully cite all the mentioned works and discuss them in detail in the related works section. Should there be more references to consider, please feel free to let us know, thanks.
>   - We would like to stress that the In-Context Discrepancy is a novel measuring score that is based on contextual prompt information, which can guide and formulate the output. But other measuring score mentioned by the reviewer is mostly based on classification probability. Therefore, our measurement is novel and different from existing works. We will clarify this in the future version.
>
> - Typo.
>   - Thanks for pointing them out, we will carefully address all of them.
>
> - Efficacy.
>   - By comparing to other well-known baseline methods, our method can further enhance the learning performance in most scenarios.
>   - Only under CIFAR-100 dataset with low noise rate, such as 20\% and 50\%, the performance of NoiseGPT shows degradation. This is because the performance of NoiseGPT highly relies on the noise detection threshold, which is set fixed as 0.72 throughout Table 3. In fact, under low noise rates, the detection threshold should be higher than that of high noise rates, in order to prevent overconfidently selecting clean examples as noisy ones. Our experimental results in section 4.4 validate such a claim, and we further provide an ablation study in Figure D in the attachment file of the author rebuttal to show that when setting proper threshold, the performance of NoiseGPT can still be effective compared to Pro-Mix.
>   - In challenging scenarios under high noise rates, NoiseGPT is quite effective in achieving 8.5\% and 18.5\% performance improvement compared to Pro-Mix under 80\% and 90\% noise rate, respectively.
>
> - Related work on Sample Cleaning.
>   - We have carefully provided the discussion of related works as mentioned in the author rebuttal.

---

> > ### Comment · Reviewer_TkfP · 2024-08-10
> > **Response to the rebuttal**
> >
> > Thanks to the authors for the rebuttal. The author mentioned that " In-Context Discrepancy is a novel".
> >
> > The author gives the experiment results as in Fig. 2, where the prediction probability distribution between the perturbed noise sample and clean samples is totally different. Therefore the author proposes the  In-Context Discrepancy to detect the noise samples.
> > However, the core idea that the prediction probability distribution between the perturbed noise sample and clean samples is totally different has been broadly studied in the robustness learning area. Many researchers study the prediction distribution between the noise samples and clean samples. However, the related works and analysis are missing. The author just claims that " In-Context Discrepancy is a novel".
> >
> > The author combines the GPT and noise label learning, which is an "A+B" work. The core idea is to find the difference between the noise sample and clean samples. However, the adopted core idea is similar to existing works. Therefore, I think the novelty of the approach is limited.

---

> > > ### Author Response · Authors · 2024-08-11
> > > **Reply**
> > >
> > > Dear Reviewer TkfP:
> > >
> > > Thanks for your quick response. We will include the works you have mentioned in the later version of our paper.
> > >
> > > Although the form and terminology of perturbations in NoiseGPT and robustness learning are similar, they are basically different. In robustness learning, perturbation methods are normally designed to attack models by subtly altering samples in ways that are imperceptible to the human eye, leading to incorrect classifications.
> > > - These methods typically introduce perturbations on the sample’s pixels by adding gradients opposite to the model’s predictions.
> > > - Existing research on model prediction distributions in robustness learning and robust training focuses on the distributional differences between samples before and after adding perturbations.
> > >
> > > In contrast, our study in NoiseGPT:
> > > - Our perturbations are applied at the latent feature level of the samples instead of a pixel level, which is a fusion of features from both query and perturbing samples.
> > > - In-Context-Discrepancy examines the different patterns of change in clean and noisy samples before and after perturbation: Clean samples exhibit prediction distribution changes similar to those caused by adversarial perturbations when information from noisy samples is incorporated, while noisy samples’ prediction scores are less susceptible to noisy perturbation. This differential response to perturbations between clean and noisy samples enables us to apply MLLMs to noise detection problems.
> > >
> > > Consequently, our approach significantly differs from previous studies on prediction distributions and introduces methods such as MoF to better address label noise problems in datasets. Ultimately, our method achieves superior performance compared to other state-of-the-art (SoTA) methods. All of these demonstrate the substantial novelty of the proposed method.
> > >
> > > Thanks again for your discussion, we hope to hear your further opinions soon.
> > >
> > > Best,
> > > Authors.

---

> ### Author Response · Authors · 2024-08-10
> **Further discussion**
>
> Dear Reviewer TkfP:
>
> We really appreciate your efforts to help improve this paper. We have carefully addressed the mentioned concerns, such as citations of relevant works, and the limitation of improvement in classification performance compared to Pro-Mix. Experiments have also been added to elaborate on these problems.
>
> Having further discussions really helps to achieve consensus and clarify misunderstandings, we are eager to know if there are any remaining problems. We will try our best to address them.
>
> Best,
> Authors.

---

### Author Rebuttal · Authors · 2024-08-07

We thank all reviewers for reading and highlighting our paper, including
- 1)”The paper presents a new application of MLLMs for label noise detection and rectification, introducing the concept of probability curvature…” (R1);
- 2)”The observation about the sensitivity of noisy samples to noise in the latent space is intuitive and motivates the approach well. NoiseGPT going beyond simply prompting the MLM to answer whether the label is clean or noisy strengthens the contribution.” (R2);
- 3)”Experiments were carried out on various synthetic noise and human annotated noise data sets to demonstrate the performance of NoisyGPT…” (R3).

They also voiced some valid concerns and put forward some constructive suggestions, which we will further elaborate on below.

- R4 questioned the novelty, R1 mentioned the similarity between probability curvature in DNNs and ICD, and R3 questioned that NoiseGPT as well as previous works leverage the memorization effects.
  - The utilization of probability curvatures of DNN are behaviors of loss function during training where clean examples tend to have a lower loss, as indicated by memorization effects. However, our method leverages the inherent optimization in MLLMs to associate visual features with corresponding text labels in a zero-shot manner. Thus, the ICD curvature is essentially different from existing probability curvatures and can be proposed as a novel method to mitigate label noise problems.

- R2 and R3 indicated the completeness of performance comparison with CLIP and smaller MLLM model.
  - To this end, we make experiments respectively with CLIP as label corrector and FLAN-T5-XL as LLM backbone of smaller size. Results are shown in Table A.

- R1 and R4 put forward that some relevant works should be mentioned and discussed. These works will be added to the discussion in related works.
  - Existing LNL methods can be categorized into three types, data cleaning, loss-adjustment-based approaches, and sample-selection-based approaches. Data cleaning endeavors to filter out examples whose labels are likely to be corrupted [1,2]. Previous works in this branch leverage various methods [3,4,5] such as bagging, boosting, K-nearest neighbor, and anomaly detection to exclude falsely labeled instances. Nonetheless, these methods tend to over-clean samples that are even true-labeled, resulting in aggravation of shortage of data in many cases. Changes in probability curvatures of DNNs [6,7] during training based on memorization effects are also utilized to filter noisy examples. However, their robustness is strongly correlative to the training setting. Besides, CLIP[9] as a zero-shot classifier is leveraged to rectify noise, which cannot still distinguish noisy examples apart.

- R4 mentioned limited performance improvement of NoiseGPT compared to baseline on CIFAR-100 sym. 20%/50%.
  - The performance is degraded because the performance of NoiseGPT relies highly on the detection threshold, which is set fixed as 0.72 throughout Table 3. In fact, under low noise rates, the detection threshold should be higher than that of high noise rates, in order to prevent overconfidently selecting clean examples as noisy ones. Our experimental results in section 4.4 validate such a claim, and we further provide an ablation study in Figure D in the attachment to show that when setting proper threshold, the performance of NoiseGPT can still be effective compared to Pro-Mix.

- R1 indicated further exploration into the sensitivity of hyperparameters.
  - We add an experiment illustrating the noise rectification performance under different perturbation numbers $n$ in Figure A. The computing time of NoiseGPT can be reduced by selecting a smaller hyperparameter while maintaining the performance.
  - We demonstrate the influence of threshold τ on classification performance on two datasets in Figure D.

- R2 questioned the complementarity between NoiseGPT and baselines.
  - Since baselines like Pro-Mix and DivideMix have noise detection modules of their own, we make a comparison of noise detection in Table B to demonstrate the superiority of NoiseGPT in noise detection and its effectiveness in enhancing the classification performance of baselines.

- R2 questioned the utilization of WebVision dataset in our experiment settings and the performance of NoiseGPT on datasets that MLLM backbones are trained on.
  - We understand that it would be more relevant to use the training set of Webvision which contains label noise crawled from the Internet. However, it would be difficult to quantitively analyze the performance of detection and rectification without ground-truth labels as reference. Thus, to demonstrate the capability of NoiseGPT to filter out noisy examples within a web dataset, we visualize the distribution of ICD scores from mini-WebVision, as illustrated in Figure B. The overall distribution is similar to what is shown in Figure 3, indicating the separation of clean and noisy examples.

[1] Wheway V. Using boosting to detect noisy data[C]
[2] Sluban B, Gamberger D, Lavrač N. Ensemble-based noise detection: noise ranking and visual performance evaluation[J].
[3] Delany S J, Segata N, Mac Namee B. Profiling instances in noise reduction[J].
[4] Gamberger D, Lavrac N, Dzeroski S. Noise detection and elimination in data preprocessing: experiments in medical domains[J].
[5] Thongkam J, Xu G, Zhang Y, et al. Support vector machine for outlier detection in breast cancer survivability prediction[C].
[6] Garg I, Ravikumar D, Roy K. Memorization through the lens of curvature of loss function around samples[J].
[7] Ravikumar D, Soufleri E, Hashemi A, et al. Unveiling privacy, memorization, and input curvature links[J].
[8] Liu S, Niles-Weed J, Razavian N, et al. Early-learning regularization prevents memorization of noisy labels[J].
[9] Liang C, Zhu L, Shi H, et al. Combating Label Noise With A General Surrogate Model For Sample Selection[J].

---

> ### Comment · Area_Chair_nDTv · 2024-08-08
> **Dear reviewers -- please read the rebuttal and respond**
>
> During August 7-13 there will be an open discussion between the reviewers and authors. I observed that the author responded to reviewer questions elaboratively, and Reviewer 81A3 have already responded to the rebuttal (thank you!).
>
> Reviewer TkfP, gxfG, QCqA --- Please use this opportunity to further clarify remaining concerns and confusions.
>
> Thank you!
> AC

---

### Author Response · Authors · 2024-08-14
**Rebuttal Summary**

Dear Reviewers, AC, and SAC:

We deeply thank the hard work done by AC and SAC such as assigning reviewers, guiding the process, and further organizing the discussion. We also sincerely appreciate the reviewers for taking the time to read our paper, provide constructive opinions, and get involved in our discussion. Without your elaborative help, our paper could not have been improved.

Here we summarize our rebuttal to present a macro perspective which could hopefully help grasp our contribution and modification quickly.

Through interactive discussion, several consensuses have been achieved:
- This paper is clearly formulated and well-written.
- Most reviewers found the idea of our paper interesting and potential to be widely adopted.

Although several reviewers questioned the complementarity of performance comparison between NoiseGPT and other SoTA methods, we have carefully provided more results on the WebVision dataset, demonstrating the extendable effectiveness of NoiseGPT.
| Dataset\Method | ELR | Mix | DivideMix | NoiseGPT+DivideMix |
| ----------- | ----------- | ----------- | ----------- | ----------- |
| Webvision | 73.00 | 74.96 | 77.32 | 78.10 |

For a fair comparison, we evaluate the noise detection performance of NoiseGPT and Pro-Mix with precision, recall, and F1 score on the same dataset setting. NoiseGPT gains better results on this metrics.
| Method\Metrics | Precision | Recall | F1 |
| ----------- | ----------- | ----------- | ----------- |
| DivideMix | 94.36% | 92.63% | 93.49% |
| Pro-Mix | 96.59% | 93.78% | 94.57% |
| NoiseGPT | 97.80% | 94.39% | 96.07% |

Although two reviewers remained silent during the discussion phase, here we stress our contribution for efficient evaluations:
- Investigate the properties of MLLM output possibilities and propose In-Context Discrepancy framework to combat label noise in datasets, which examines the different patterns of change in clean and noisy samples before and after perturbation. ICD goes further beyond the previous works because it:
  - Does not require the training of DNNs as previous LNL works do, which are based on memorization effects.
  - Addresses the varying output distributions of different categories when MLLMs are simply prompted to produce a “True” or “False” possibility. Thus, clean and noisy samples can be separated by a unique threshold.
- Carefully leverage these properties by designing a MoF method to introduce perturbations into query samples.
  - Our MoF fuses information of query samples and perturbing samples on the latent representation level, which is essentially different from previous perturbation methods used in adversarial learning.
- Possesses superior capability of detecting and rectifying noisy labels compared with previous LNL works such as Pro-Mix, DivideMix, and M-correction.
  - Since several reviewers mentioned the biases among different categories in detection and rectification. We conduct experiments on the same dataset with NoiseGPT and Pro-Mix, which can be seen in the table below. Our method shows a much lower variance, demonstrating a smaller bias.
| Method\Class index| 0 | 1 | 2 | 3 | 4 | 5 | 6 | 7 | 8 | 9 | Variance |
| ----------- | ----------- | ----------- | ----------- | ----------- | ----------- | ----------- | ----------- | ----------- | ----------- |----------- |----------- |
| Pro-Mix | 9.79% | 11.02% | 9.52% | 7.84% | 10.05% | 8.23% | 10.61% | 11.01% | 10.80% | 11.12% | **1.24** |
| NoiseGPT | 10.43% | 10.67% | 8.95% | 10.63% | 10.91% | 9.04% | 10.75% | 8.46% | 9.98% | 10.18% | **0.68** |
  - Additional experiments are made with a different LLM backbone FLAN-T5-XL, and CLIP as a label corrector.
| Method\Dataset | C10sym20 | C10sym50 | C10sym80 | C10sym90 | C10asym40 | C100sym20 | C100sym50 | C100sym80 | C100sym90 |
| ----------- | ----------- | ----------- | ----------- | ----------- | ----------- |  ----------- |  ----------- |  ----------- |  ----------- |
| CLIP-ViT-L | 10.2% | 18.7% | 24.4% | 29.6% | 16.1% | 18.5% | 30.8% | 44.2% | 45.4% |
| FLAN-T5-XL | 9.8% | 18.1% | 25.7% | 31.2% | 11.6% | 18.0% | 31.4% | 47.9% | 50.3% |
| FLAN-T5-XXL| **7.4%** | **13.6%** | **19.3%** | **24.4%** | **9.4%** | **16.1%** | **28.0%** | **40.2%** | **44.6%** |
  - Ablation studies are conducted with different hyper-parameter settings.

In the past several months, we have tried our best to improve the quality of this paper and address each concern from all reviewers. We sincerely hope our effort could be contributed to the community and advance the development of LNL. Thanks again for your kind help and constructive opinions, we will keep trying our best to polish this paper in the future.

Sincerely,

Authors.

---

### Decision · Program_Chairs · 2024-09-25

**Decision:**

Accept (poster)

**Comment:**

This manuscript received mixed reviews both pre- and post-rebuttal. After the rebuttal, the AC initiated and discussed with the reviewers. It seems two key concerns remain after the rebuttal, author-reviewer discussion, and post-rebuttal discussion.

1. Reviewer TkfP complains about the novelty, and considers the proposed method as "A+B". During post-rebuttal discussion, reviewer TkfP listed quite some references to support this claim. The AC read through the list and many posts by the authors and by reviewer TkfP. After careful evaluations, the AC feels that the list of references are talking about the theoretical aspects of noise rectification, which does not fall in the scope of this discussion or the purpose of NoiseGPT (this submission). Hence, the AC is with the authors in terms of this disagreement.

2. Reviewer 81A3 was still concerned about WebVision style experiments. The author(s) did provided some experimental results, but they were considered (partly) inappropriate by reviewer 81A3. After reading the materials, the AC is with reviewer 81A3 in this issue. The authors are suggested to conduct experiments following the adivce of reviewer 81A3, and incorprate these results into the paper.